# Perturbation of Autophagy by a Beclin 1-Targeting Stapled Peptide Induces Mitochondria Stress and Inhibits Proliferation of Pancreatic Cancer Cells

**DOI:** 10.3390/cancers15030953

**Published:** 2023-02-02

**Authors:** Na Li, Xiaozhe Zhang, Jingyi Chen, Shan Gao, Lei Wang, Yanxiang Zhao

**Affiliations:** 1Shenzhen Research Institute, The Hong Kong Polytechnic University, Shenzhen 518057, China; 2State Key Laboratory of Chemical Biology and Drug Discovery, Department of Applied Biology and Chemical Technology, The Hong Kong Polytechnic University, Hung Hom, Kowloon, Hong Kong 999077, China

**Keywords:** PDAC, EGFR, autophagy, stapled peptide

## Abstract

**Simple Summary:**

Autophagy has long been regarded as playing both pro- and anti-proliferative roles in many types of cancer, including PDAC. As the autophagy inhibitor CQ failed to show therapeutic efficacy in PDAC clinical trials, it is worth exploring whether the alternative approach, i.e., elevation of the autophagic activity, can exert an anti-tumor effect. The aim of our study was to assess whether a Beclin 1-targeting stapled peptide can exert an anti-proliferative effect in PDAC by perturbing the already elevated autophagy process. Our study reports for the first time that the Beclin 1-targeting stapled peptide Tat-SP4 potently inhibited the proliferation of PDAC cells through the combined effect of excessive autophagy, enhanced endolysosomal degradation of EGFR and significant mitochondria stress. Tat-SP4 induced non-apoptotic cell death in PDAC cells, which is in distinct contrast to apoptosis induced by CQ. In summary, the perturbation of the autophagy process by Tat-SP4 may serve as a novel therapeutic for PDAC.

**Abstract:**

Pancreatic ductal adenocarcinoma (PDAC) is the most common type of pancreatic cancer, with a dismal five-year survival rate of less than 10%. PDAC possesses prominent genetic alterations in the oncogene KRAS and tumor suppressors p53, SMAD4 and CDKN2A. However, efforts to develop targeted drugs against these molecules have not been successful, and novel therapeutic modalities for PDAC treatment are urgently needed. Autophagy is an evolutionarily conserved self-degradative process that turns over intracellular components in a lysosome-dependent manner. The role of autophagy in PDAC is complicated and context-dependent. Elevated basal autophagy activity has been detected in multiple human PDAC cell lines and primary tumors resected from patients. However, clinical trials using chloroquine (CQ) to inhibit autophagy failed to show therapeutic efficacy. Here we show that a Beclin 1-targeting stapled peptide (Tat-SP4) developed in our lab further enhanced autophagy in multiple PDAC cell lines possessing high basal autophagy activity. Tat-SP4 also triggered faster endolysosomal degradation of EGFR and induced significant mitochondria stress as evidenced by partial loss of Δψ, increased level of ROS and reduced OXPHOS activity. Tat-SP4 exerted a potent anti-proliferative effect in PDAC cell lines in vitro and prohibited xenograft tumor growth in vivo. Intriguingly, excessive autophagy has been reported to trigger a unique form of cell death termed autosis. Tat-SP4 does induce autosis-like features in PDAC cells, including mitochondria stress and non-apoptotic cell death. Overall, our study suggests that autophagy perturbation by a Beclin 1-targeting peptide and the resulting autosis may offer a new strategy for PDAC drug discovery.

## 1. Introduction

Pancreatic ductal adenocarcinoma (PDAC) is the most common type of pancreatic cancer and also among the deadliest in all cancer types, with a dismal five-year survival rate of ~10% [1,2,3]. A major reason for this poor prognosis is the lack of specific symptoms at the early stage of tumor growth. As a result, the majority of cases are diagnosed at advanced or metastatic stages when curative surgery is not applicable [1,2,3]. Therapeutic options for PDAC are limited, with conventional radiotherapy and systemic chemotherapy being the two most frequently prescribed modalities [2]. Notably, combination chemotherapy such as FOLFIRINOX consisting of four chemo drugs or gemcitabine combined with nab-paclitaxel offers a significant survival benefit of ~3–6 months [4,5]. However, the recurrence rate is high, with ~60–70% of patients suffering relapse within two years and eventually dying of their disease [3]. Furthermore, while two targeted drugs, erlotinib and olaparib, were approved for PDAC, they either showed insufficient clinical efficacy (erlotinib) or only benefits ~5–9% of patients carrying specific germline BRCA mutations (olaparib) [6]. Furthermore, PDAC is known for its low mutational burden and immunosuppressive microenvironment. As a result, while immune checkpoint inhibitors like anti-PD1 or anti-PD-L1 antibodies have shown clinical success in many cancer types when combined with chemotherapy, their efficacy in PDAC is yet to be confirmed [6]. In the absence of more effective treatment options, PDAC is predicted to become the second leading cause of cancer-related deaths in developed countries such as the U.S. within the next decade [1,2,3].

Autophagy is an evolutionarily conserved self-degradative process that turns over intracellular components in a lysosome-dependent manner [7,8,9]. A hallmark feature of autophagy is the formation of double-membraned autophagosomes to engulf cytosolic content in bulk and eventually fuse with lysosomes to degrade and recycle the sequestered cargo [7,8,9]. Autophagy usually operates at a basal level for the maintenance of cellular homeostasis but can be significantly up-regulated under stress conditions such as nutrient deprivation and hypoxia to scavenge metabolites and promote survival [7,8,9]. The role of autophagy in PDAC is complicated and context-dependent. Under the nutrient-depleted tumor microenvironment, autophagy has been proposed as a pro-survival mechanism for PDAC cells to sustain their fast proliferation [10]. Indeed, elevated basal autophagy activity has been detected in multiple human PDAC cell lines and primary tumors resected from patients [11]. Chloroquine (CQ), an inhibition of autophagy, showed stronger anti-proliferative efficacy in autophagy-elevated 8988T PDAC cells than in the H460 lung cancer cells with low basal autophagy [11]. A study using a mouse model for PDAC also showed that autophagy inhibition in the pancreas by Atg5 knockdown reduced invasive cancer and prolonged survival [12]. However, a study by Eng et al. showed that autophagy was dispensable for PDAC proliferation because CQ showed comparable anti-proliferative efficacy in wild-type and Atg7-knockout Panc 10.05 cells [13]. Furthermore, a clinical trial testing autophagy inhibitor hydroxychloroquine (HCQ) in patients with metastatic PDAC failed to show therapeutic efficacy [14]. Overall, while autophagy is intimately involved in PDAC, how to target this process for effective drug discovery remains to be investigated.

Beclin 1 is a core member of the Class III phosphatidylinositol-3-kinase (PI3KC3) complex and is responsible for recruiting two positive modulators, Atg14L and UVRAG, in a mutually exclusive manner to form Atg14L- or UVRAG-containing PI3KC3 complex with enhanced lipid kinase activity [15,16,17,18]. As a result, Beclin 1 is essential for multiple PI3KC3-mediated cellular processes, including autophagy, endolysosomal trafficking and phagocytosis [16,19,20,21]. Beclin 1 is also a haploinsufficient tumor suppressor and mono-allelically deleted in 40–75% of human sporadic breast cancer and ovarian cancer [19,20,22]. Intriguingly, Beclin 1 is highly expressed in clinical samples of PDAC, although its relationship to prognosis and survival is not clear [23,24]. We reason this unique feature could be exploited to modulate autophagy in PDAC for therapeutic benefit.

In our previous studies, we determined the crystal structure of Beclin 1 coiled-coil domain in the inactive homodimer state and its complex with UVRAG in the active heterodimer form [25,26]. Guided by these structures, we designed Beclin 1-targeting stapled peptides that bound to the Beclin 1 coiled-coil domain with high affinity to reduce its homodimerization and to promote the Beclin 1-Atg14L/UVRAG interaction [26,27]. All these peptides contain two distinct segments, including the N-terminal Tat sequence for cell penetration and the C-terminal Beclin 1-targeting segment modified with a hydrocarbon staple with either (i, i + 7) or (i, i + 4) linkage to stabilize the α-helical structure (Figure 1A). One lead peptide, Tat-SP4, was shown to promote autophagy and endolysosomal degradation of epidermal growth factor receptor (EGFR) in a panel of cell lines [26]. Tat-SP4 also synergized with erlotinib to potently inhibit the proliferation of non-small-cell lung cancer cell lines A549 and H1975 by degrading the over-expressed EGFR and inducing non-apoptotic cell death [28].

Here, we report that Tat-SP4 further enhanced autophagy in multiple PDAC cell lines that have been reported to possess high basal autophagy activity. Tat-SP4 also promoted endolysosomal degradation of EGFR, an oncogenic receptor over-expressed in these cell lines, and exerted an anti-proliferative effect with micromolar potency. Intriguingly, Tat-SP4 treatment led to mitochondria stress and non-apoptotic cell death. In a xenograft model of PDAC, Tat-SP4 effectively inhibited tumor growth with no overt toxicity. These data suggest that the perturbation of autophagy with Beclin 1-targeting stapled peptides may serve as a novel anti-proliferative strategy for PDAC.

## 2. Materials and Methods

### 2.1. Reagents and Antibodies

Chloroquine (CQ; Sigma-Aldrich, St. Louis, MO, USA), epidermal growth factor (EGF; Gibco, Waltham, MA, USA), EDTA-free protease inhibitor cocktail (Roche, Basel, Switzerland), trypsin (Invitrogen, Waltham, MA, USA), trypan blue (Gibco), oligomycin (Cayman, Ann Arbour, MI, USA), FCCP (Cayman), rotenone (Sigma-Aldrich), antimycin A (Sigma-Aldrich), matrigel (Corning, Corning, NY, USA). anti-β-Actin antibody (Santa Cruz, CA, USA), anti-LC3 antibody (Abnova, Taipei, Taiwan), anti-p62 antibody (Abnova), anti-EGFR antibody (Santa Cruz), anti-Mouse IgG-HRP (Sigma-Aldrich), anti-Rabbit IgG-HRP (Sigma-Aldrich).

### 2.2. Chemical Synthesis of Stapled Peptides

The Tat-SP4 stapled peptide was purchased from GL Biochem (Shanghai, China) Ltd. The synthesis process was the same as reported in our previous study [27]. Briefly, the peptide was synthesized by automated solid-phase method with olefin-containing amino acids incorporated at the designated positions. The hydrocarbon staple was formed on olefin-containing amino acids by ring-closing metathesis reaction using the Grubbs catalyst. Chemical structure and purity of the final product were characterized by HRMS and HPLC. Purity of each synthesized peptide is >95%. Stock solution of each obtained peptide was prepared by dissolving the sample in pure water to a concentration of 20 mM. 

### 2.3. Cell Lines and Cell Culture

The pancreatic ductal adenocarcinoma (PDAC) cell lines PANC-1 and MIA PaCa-2 were purchased from the American Type Culture Collection (Manassas, VA, USA), and BxPC-3 cell was purchased from Stem Cell Bank, Chinese Academy of Sciences, Shang Hai, China. These three PDAC cells were cultured in RPMI 1640 with 10% (*v*/*v*) FBS at 37  °C in a 5% CO_2_ incubator. All cell lines used in the experiments were mycoplasma detected negative by MycoAlertTM PLUS Mycoplasma Detection Kit before and during the experiment.

### 2.4. Cell Viability Assay

Trypan blue exclusion assay was used to measure the cell viability. PDAC cells were seeded in 96-well plates with 2 × 10^4^ cells per well for 12 h before being treated with indicated concentrations of Tat-SP4 for 24 h. The number of viable cells was counted manually, and the IC_50_ value was calculated from the curve fitted to concentration–response datasets. All experiments were repeated in triplicate, and the mean was calculated.

### 2.5. 5-Day Cell Growth Curve

The long-term effect of Tat-SP4 in inhibiting cell proliferation was assessed by plotting a five-day cell growth curve. PDAC cells were seeded in 24-well with 4 × 10^4^ cells per well for 12 h before being treated with indicated concentrations of Tat-SP4. Treated cells were then cultured for 5 days, and the number of viable cells was counted every 24 h by trypan blue exclusion assay. The viable cell number is plotted on a graph to create a growth curve.

### 2.6. Immunoblot Analysis

Cells were washed with phosphate buffered saline (PBS) buffer and lysed by extraction buffer (62.5 mM Tris-HCl, pH 6.8, 2% SDS, 25% glycerol, 5% β-mercaptoethanol) with freshly added protease inhibitor cocktail (Roche). The whole cell lysate was then loaded in SDS-PAGE and analyzed by Western blot.

### 2.7. EGFR Degradation Assay

PDAC cells in 6-well plate were washed with PBS two times and cultured in serum-free medium for 12 h. EGF (200 ng/mL) was then added to the medium to induce the endocytosis of EGFR. Cells were collected at the indicated time point after EGF stimulation and lysed for immunoblot analysis.

### 2.8. Flow Cytometry

Flow cytometry (BD Accuri C6) was employed to probe the mitochondrial membrane potential (MMP), cellular reactive oxygen species (ROS) level and the degree of cellular apoptosis in PDAC cells, and all data were analyzed by BD Accuri C6 software. To measure the MMP, the PDAC cells in 6-well plates were stained with 10 nM tetramethylrhodamine (TMRM) for 30 min at 37 °C, followed by washing with PBS and digesting with trypsin for flow cytometry. For the quantitation of ROS in PDAC cells, 20 μM 2′,7′ –dichlorofluorescin diacetate (DCFDA), the cell-permeant reagent was used to stain the cells for 30 min at 37 °C. The degree of cellular apoptosis and necrosis was assessed by labeling the PDAC cells with Annexin V and PI. Typically, cellular population in the lower left quadrant is live cells (Annexin V−, PI−), the lower right quadrant represents early apoptotic cells (Annexin V+, PI−), the upper right quadrant represents late apoptotic cells (Annexin V+, PI+), the upper left quadrant represents necrotic cells (Annexin V−, PI+).

### 2.9. Seahorse Analysis of OXPHOS

Seahorse XFe24 analyzer (Agilent, Santa Clara, USA) was used to continuously monitor the oxygen consumption rate (OCR) of PDAC cells. Briefly, 6 × 10^4^ BxPC-3 cells were plated in XFe24 cell culture plate in RPMI 1640 media supplemented with 10% FBS and 1% PS one day prior to the experiment. Before the measurement, the medium was replaced with Seahorse medium supplemented with 2 mM L-glutamine, 1 mM sodium pyruvate and 10 mM glucose. Altogether, 1 μM oligomycin, 1 μM carbonyl cyanide-4-phenylhydrazone (FCCP), 0.5 μM antimycin A and 0.5 μM rotenone were used as inhibitors according to the manufacturer’s instructions to detect the changes of OCR in BxPC-3 cells.

### 2.10. Animal Study

Female BABL/c nude mice, 6 weeks old, were obtained from Beijing Vital-River Lab Animal Technology Co., Ltd., (SCXK JING 2007-0001). To establish the subcutaneous implantation models, we subcutaneously injected 5 × 10^6^ Bxpc-3 cells into the flanks of the mice. Tumor volumes and the body weights of mice were measured regularly. The mice were randomly grouped when the average tumor volumes reached around 100 mm^3^ and then were treated with PBS or Tat-SP4 (40 mg/kg/day) via intraperitoneal injection daily (*n* = 6 in each group). The mice were sacrificed at a time-defined endpoint, the tumors were weighed and measured, and the major organs of mice were collected.

### 2.11. Statistical Analyses

Results were presented as mean ± SEM. Statistical significance was assessed by either two-tailed, unpaired Student’s *t*-test or ordinary one-way ANOVA for multiple cohorts (GraphPad Software). *p*-values < 0.05 were considered statistically significant.

## 3. Results

### 3.1. A Beclin 1-Targeting Stapled Peptide Tat-SP4 Induced Autophagy in Multiple PDAC Cell Lines

PDAC shows extensive genetic, cellular and microenvironment heterogeneity. A PDAC tumor contains a mixture of several distinct cell types, including pancreatic cancer cells, cancer stem cells and tumor stroma [29]. The progression from precancerous intraepithelial neoplasia to invasive PDAC depends critically on the accumulation of genetic alterations that perturb multiple oncogenic or tumor suppressor signaling pathways, with KRAS, p53, SMAD4 and CDKN2A as the most frequently mutated genes [30]. Integrated genomic and transcriptome profiling of clinical samples has categorized PDAC into two major subtypes, including the squamous type, which is basal-like with poor clinical outcome, and the pancreatic progenitor type (also called classical or epithelial type) that preferentially express genes linked to cell fate determination [31,32,33]. To assess whether the Beclin 1-targeting Tat-SP4 could perturb autophagy in PDAC cells with diverse genetic and phenotypic backgrounds, we decided to use three widely used cell lines, including BxPC-3, MIA PaCa-2 and PANC-1. These cell lines differ in terms of their genotypic status for frequently altered genes KRAS, p53, SMAD4 and CDKN2A [34]. They also show phenotypic differences in cell adhesion, angiogenic potential and tumorigenicity [34]. Furthermore, all three cell lines possess sufficient basal autophagy and thus are amenable to modulation by Tat-SP4 [11].

The design principle for our Beclin 1-targeting stapled peptides is to disrupt the functionally inactive Beclin 1 homodimer and promote the formation of functionally active Beclin 1-Atg14L/UVRAG heterodimer to induce autophagy. Thus, Tat-SP4 is not expected to affect the overall level of Beclin 1 in vivo but to promote the homodimer-to-heterodimer transition instead. This assumption was confirmed in HEK293 cells in our previous study [26]. Additionally, the formation of more autophagosomes in HEK293 cells upon Tat-SP4 treatment was observed by confocal immunofluorescence microscopy. Tat-SP4 likely induces autophagy in the same manner in all Beclin 1-expressing cell lines. Thus, it would be sufficient to assess whether Tat-SP4 could induce autophagy in these PDAC cell lines by tracking two autophagy markers LC3-II and p62. Upon autophagy induction, the microtubule-associated protein 1 light chain 3 (LC3) is converted from the unmodified form (LC3-I) to the lipidated form (LC3-II) and clusters onto the nascent autophagic membrane to promote autophagosome biogenesis [9]. Additionally, p62 is an autophagy receptor that binds to autophagic cargos and sequesters them into autophagosomes for lysosomal delivery and degradation [9]. Thus, an increase in LC3-II and a decrease in p62 are two widely used markers for cellular autophagic activity. Our data show that Tat-SP4 treatment for 24 h led to a dosage-dependent increase in LC3-II levels in BxPC-3 cells, with ~50% increase at 10 μM and ~100% increase at 20 μM (Figure 1B,C). Concomitantly, the p62 level decreased by ~20% at 10 μM and ~30% at 20 μM (Figure 1C). Similar changes in LC3-II and p62 levels were also observed in the presence of CQ, one of the most used autophagy inhibitors (Figure 1B,C). Overall, Tat-SP4 induced a potent autophagic response in BxPC-3 cells. 

Tat-SP4 also induced a similar autophagic response in MIA PaCa-2 and PANC-1 cells, as shown by dosage-dependent changes in LC3-II and p62 (Figure 1D–G). Notably, MIA PaCa-2 showed the most significant change with a >100% increase in LC3-II and >50% reduction of p62 after Tat-SP4 treatment at 20 μM (Figure 1D,E). These results confirm that Tat-SP4 further enhanced autophagic response in PDAC cell lines that have been reported to possess elevated basal autophagy activity [11].

### 3.2. Tat-SP4 Promoted Endolysosomal Degradation of EGFR in PDAC Cells

Our previous studies have shown that Tat-SP4 promotes both autophagy and endolysosomal degradation of EGFR, two processes that critically depend on the Beclin 1-mediated PI3K3C complex [26]. The EGFR signaling axis is frequently activated in PDAC to sustain excessive proliferation [35]. Additionally, endogenous EGFR at the cell membrane is subject to agonist-induced internalization into endosomes, followed by trafficking either back to the cell membrane for sustained signaling or to lysosomes for degradation [36]. To assess whether Tat-SP4 promotes endolysosomal degradation of EGFR in PDAC, we first treated the three PDAC cell lines with EGF to assess their endocytic trafficking pattern of endogenous EGFR. Our data show that the EGFR level remained steady over a period of 6–12 h post-EGF treatment, thus suggesting that the endogenous EGFR is largely recycled back to the cell membrane after EGF-induced internalization (Figure 2A–F). Tat-SP4 treatment at 20–30 μM induced noticeable EGFR degradation in all three PDAC cell lines, with >50% reduction at 6–12 h post-EGF treatment (Figure 2A–F). These results suggest that Tat-SP4 can alter the endocytic trafficking pattern of EGFR in PDAC cells and promote its endolysosomal degradation.

### 3.3. Tat-SP4 Exerted Potent Anti-Proliferative Effect on PDAC Cell Lines

In our previous studies, Tat-SP4 showed moderate anti-proliferative efficacy in cell line models of non-small-cell lung cancer (NSCLC) and HER2+ breast cancer with IC_50_ values of ~30–50 μM [27,28]. To assess whether Tat-SP4 has a similar effect on PDAC, we used the trypan blue exclusion assay to measure its IC_50_ values for the three cell lines. Our data show that Tat-SP4 exerted a stronger anti-proliferative effect on PDAC cell lines with IC_50_ of ~12–20 μM (Figure 3A). A five-day proliferation assay confirmed that Tat-SP4 reduced the proliferation of BxPC-3 and MIA PaCa-2 cells by ~30% and ~80%, respectively at 15 μM, a dosage close to their respective IC_50_ values (Figure 3B,C). In comparison, Tat-SC4, a control peptide with a scrambled sequence of Tat-SP4 and no hydrocarbon stapling, showed no such effect (Figure 3B,C). Tat-SP4 also inhibited the proliferation of PANC-1 by ~50% at 30 μM, a higher dosage set to match its higher IC_50_ value (Figure 3D). These results confirm that Tat-SP4 exerts a potent anti-proliferative effect on PDAC cells.

### 3.4. Tat-SP4 Triggers Predominantly Non-Apoptotic Cell Death in PDAC Cells

Our previous study reported that Beclin 1-targeting stapled peptides, including Tat-SP4 and another similar peptide with (i, i + 4) linkage for the hydrocarbon staple induced non-apoptotic cell death in NSCLC, breast cancer cells and hepatocellular carcinoma (HCC) cells [27,28,37]. We set out to investigate if Tat-SP4 exerted a similar effect on PDAC cells. Flow cytometry measurements with Annexin V and propidium iodide (PI) staining revealed that Tat-SP4 induced predominantly non-apoptotic cell death in BxPC-3, MIA PaCa-2 and PANC-1 cells (Figure 4). For BxPC-3, treatment with 20 μM Tat-SP4 for 2 h led to ~50% non-apoptotic cell death that stained positive for both Annexin V and PI, while this number rose up to ~63% after 5 h (Figure 4A). Tat-SP4 showed weaker potency in MIA PaCa-2 and PANC-1 cells, with non-apoptotic cell death reaching ~20–30% and ~40–60% at 2 and 5 h, respectively (Figure 4B,C). In comparison, the level of apoptotic cell death that stained positive for Annexin V but negative for PI remained low at ~6–8% and steady over 5 h for all PDAC cells (Figure 4). Overall, our data confirm that Tat-SP4 induced predominantly non-apoptotic cell death in PDAC cells similar to HER2+ breast cancer cells.

### 3.5. Tat-SP4 Induced Mitochondria Stress and Impaired Oxidative Phosphorylation Activity

PDAC cells depend on elevated autophagy to salvage nutrients for anabolism and energy production through oxidative phosphorylation (OXPHOS) in mitochondria [38]. As Tat-SP4 further enhanced the already elevated autophagy activity in PDAC cells, we wondered whether such perturbation would be detrimental to mitochondria function. Flow cytometry analysis of all three PDAC cell lines after Tat-SP4 treatment showed that the subpopulation of cells with complete loss of mitochondria membrane potential (Δψ) increased in a dosage-dependent manner (Figure 5A). Similarly, Tat-SP4 also progressively increased the level of cellular ROS over the period of 2 h (Figure 5B). Notably, BxPC-3 was more susceptible to Tat-SP4 than MIA PaCa-2 or PANC-1 in terms of loss of Δψ and increase of ROS (Figure 5A,B). We then used the Agilent Seahorse XF Analyzer to measure the cellular bioenergetic profile of BxPC-3 cells. The basal oxygen consumption rate (OCR), which measures the intrinsic mitochondria-mediated OXPHOS activity, showed a small increase upon the addition of Tat-SP4 (Figure 5C,D). We also measured the maximal OCR, which reflects the unrestrained capacity of the electron transfer chain (ETC) after inhibition of ATP synthesis by oligomycin, followed by dissipation of Δψ by CCCP. Our data show that Tat-SP4 triggered a significant dosage-dependent reduction of the maximal OCR in BxCP-3 cells (Figure 5C,E). In summary, our data suggest that Tat-SP4 exerts a dichotomous effect on mitochondria. On the one hand, Tat-SP4 slightly boosts OXPHOS in functional mitochondria, probably to turn over the nutrients salvaged by the elevated autophagy activity. On the other hand, Tat-SP4 can be detrimental to mitochondria with compromised coupling between ETC and ATP synthesis. Under such context, Tat-SP4 induces mitochondria stress and impairs OXPHOS activity.

### 3.6. Tat-SP4 Inhibited PDAC Tumor Growth in a Xenograft Model

As our in vitro studies validated Tat-SP4 as a potent inducer of mitochondria dysfunction and non-apoptotic cell death in PDAC cells, we then proceeded to assess the anti-proliferative efficacy of Tat-SP4 in vivo using a xenograft model. BxPC-3 was chosen for this study due to its high sensitivity to Tat-SP4 and its tumorigenicity. BxPC-3 cells were implanted subcutaneously on 6-week female nude mice, and daily intraperitoneal injection (i.p.) of Tat-SP4 at 40 mg/kg started once tumor volume reached 100 mm^3^. Compared to the control group with a daily injection of PBS, Tat-SP4 reduced tumor volume by ~50% after 46 days of treatment (Figure 6A,B). The tumor weight was reduced by ~30% as well (Figure 6C). Additionally, Tat-SP4 treatment at 40 mg/kg showed no obvious toxicity to mice as measured by body weight and vital organs (Figure 6D,E). Whether Tat-SP4 affects the levels of LC3 and p62 in these xenograft tumor tissue shall be explored in future studies.

## 4. Discussion

PDAC possesses several prominent genetic alterations, including gain-of-function mutations in the driver oncogene KRAS and inactivating mutations in tumor suppressors p53, SMAD4 and CDKN2A [30]. While these driver alterations can serve as actionable therapeutic targets, nearly all of these molecules have been termed “undruggable” because it is very challenging to develop specific and potent modalities against them. Notably, KRAS mutation is present in more than 90% of PDAC cases, with G12 as the most frequently altered hotspot [39]. Decades of intensive drug discovery efforts have finally yielded a handful of small-molecule drugs that specifically inhibit the KRASG12C mutant by forming a covalent adduct with the mutated cysteine residue [40,41,42]. Two top candidates, adagrasib and sotorasib, were recently approved for advanced or metastatic NSCLC-carrying KRASG12C mutation, while their efficacy in PDAC is expected to be confirmed soon [43,44]. However, the frequency of KRASG12C mutation in PDAC is only ~3%, with the other more frequent mutation types, such as KRASG12D, KRASG12R and KRASG12V, still untractable [39].

PDAC also undergoes extensive metabolic reprogramming to sustain excessive proliferation in a nutrient-deficient microenvironment. Elevated autophagy, increased glycolytic flux and utilization of glutamine as the alternative carbon source are some of the most studied metabolic hallmarks of PDAC [38]. Autophagy has also been implicated in the generation, differentiation, plasticity and migration/invasion of cancer stem cells [45]. While these pathways have been explored as potential therapeutic targets, only the autophagy inhibitor HCQ reached the clinical stage and showed no therapeutic effect in PDAC patients in a phase II trial [14]. Multiple factors could have contributed to this negative outcome, such as inconsistent potency of HCQ in terms of inhibiting autophagy in vivo and crosstalk between metabolic pathways to circumvent the HCQ effect [14]. Indeed, recent studies showed that combined inhibition of autophagy and the RAF-MEK-ERK signaling pathway downstream of KRAS led to synergistic anti-proliferative effect in both PDAC cell lines in vitro and mice xenograft models in vivo [46,47]. The clinical efficacy of this combination approach will need to be validated in patient trials.

Here, we present a different approach to targeting autophagy to inhibit PDAC proliferation. Instead of reducing cellular autophagic flux, we used a Beclin 1-targeting stapled peptide Tat-SP4 to promote this process. Our data show that Tat-SP4 treatment induced robust autophagic response in a panel of PDAC cell lines on top of their high basal autophagy level. Such an increase in autophagy did not lead to faster cell proliferation, possibly because the excessive autophagic activity may perturb the metabolic homeostasis in PDAC cells. In addition to affecting autophagy, Tat-SP4 also triggered faster endolysosomal degradation of EGFR and caused significant mitochondria stress, with partial loss of Δψ, an increased level of ROS and reduced OXPHOS activity. While PDAC cells are known to rely on elevated glycolysis for energy metabolism, mitochondria activity is still indispensable because the TCA cycle supplies important intermediates for biosynthetic pathways to support fast proliferation [48]. In summary, Tat-SP4 potently inhibited the proliferation of PDAC cells through the combined effect of excessive autophagy, enhanced endolysosomal degradation of EGFR and significant mitochondria stress.

Notably, Tat-SP4 induced non-apoptotic cell death in PDAC cells, which is in distinct contrast to apoptosis induced by pharmacological inhibition of autophagy by CQ or genetic ablation of key autophagy genes [12]. While autophagy has been found to participate in many aspects of cell cycle regulation [49], the non-apoptotic cell death induced by Tat-SP4 occurred within hours without direct effect on the cell cycle. Our previous studies also reported similar non-apoptotic cell death in other cancer cell lines after Tat-SP4 treatment [27,28,37]. Excessive induction of autophagy has been reported to cause a unique form of cell death termed “autosis” [50]. Examples of autosis have been reported in cultured cells after prolonged starvation or treatment by a Beclin 1-derived peptide (Tat-Beclin 1), in cardiomyocytes after ischemia/reperfusion injury and in tumor cells after CAR-T therapy [50,51,52,53]. Autosis is non-apoptotic as it lacks the hallmarks of apoptosis and cannot be rescued by apoptosis inhibitors [50]. Autosis shows necrotic features such as focal rupture of the plasma membrane, but autotic cells remain attached to the substrate and do not float like necrotic cells [50]. Intriguingly, mitochondria in autotic cells exhibit morphological abnormalities, including fragmented networks and electron-dense structures [50]. Tat-SP4 treatment in PDAC cells does induce similar effects as autosis, including significant mitochondria stress and non-apoptotic cell death. The molecular mechanism of autosis is still at the early stage of the investigation. Future studies are required to further characterize possible autosis in Tat-SP4-treated cancer cells. Overall, our study suggests that autophagy perturbation by a Beclin 1-targeting peptide and the resulting autosis may offer a new strategy for PDAC drug discovery.

## 5. Conclusions

In summary, we report that Tat-SP4 enhanced the autophagy process in multiple PDAC cell lines possessing a high level of basal autophagy. Tat-SP4 also triggered faster endolysosomal degradation of EGFR and induced significant mitochondria stress as evidenced by partial loss of Δψ, increased level of ROS and reduced OXPHOS activity. Tat-SP4 exerted a potent anti-proliferative effect in PDAC cell lines in vitro and prohibited tumor growth in the PDAC xenograft model in vivo. Our study shows that perturbation of the autophagy process by our Beclin 1-targeting stapled peptide Tat-SP4 may serve as a novel therapeutic for PDAC.

## 6. Patents

Y Zhao, S Wu, W Yang, Y He, X Li, X Qiu. Hydrocarbon stapled polypeptide for enhancement of endosome lysosome degradation. United States Patent App. 20180002381.

## Figures and Tables

**Figure 1 cancers-15-00953-f001:**
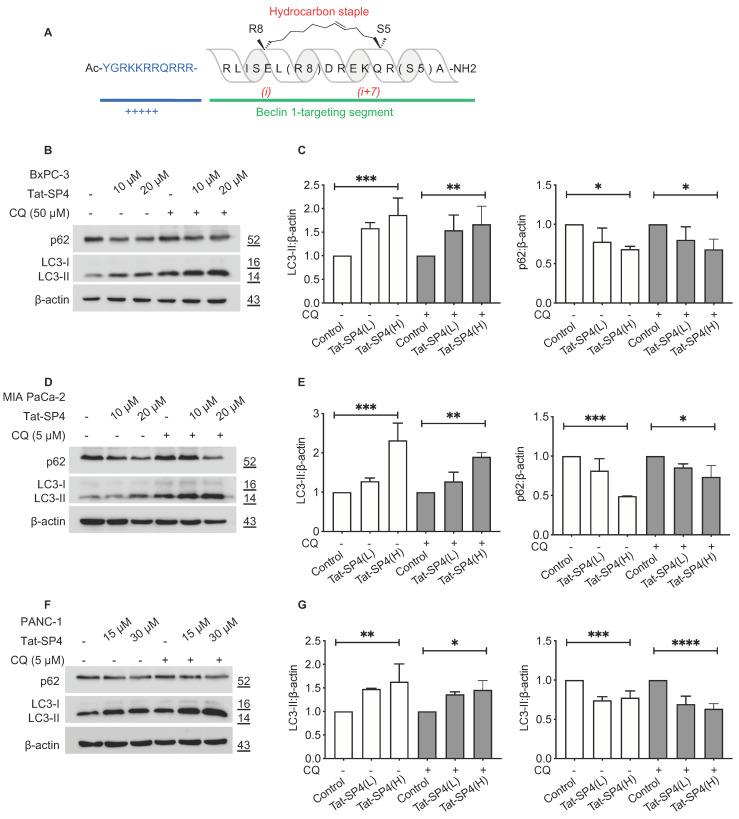
Tat-SP4 induced autophagy in multiple PDAC cell lines. (**A**) Sequence and chemical structure of Tat-SP4. A cell-penetrating Tat sequence (colored in blue) is added to the N terminal of Tat-SP4. C-terminal is the Beclin-1 targeting segment. The α-helical structure is stabilized by hydrocarbon staple. (**B**) Western blot to assess the p62 level and LC3 lipidation profile in BxPC-3 cells after treatment with 10- and 20-μM of Tat-SP4 for 24 h, in the presence or absence of CQ. (**C**) Quantification of LC3 lipidation profiles and p62 levels from the Western blot data. (**D**,**F**) Similar Western blots as (**B**), but cells are MIA PaCa-2 and PANC-1, respectively. (**E**,**G**) Quantification of LC3 and p62 levels from data in (**D**,**F**). The levels of LC3-II or p62 were normalized to the β-actin level. Data are presented as mean ± SEM (*n* = 3); * *p* < 0.05, ** *p* < 0.01, *** *p* < 0.001, **** *p* < 0.0001.

**Figure 2 cancers-15-00953-f002:**
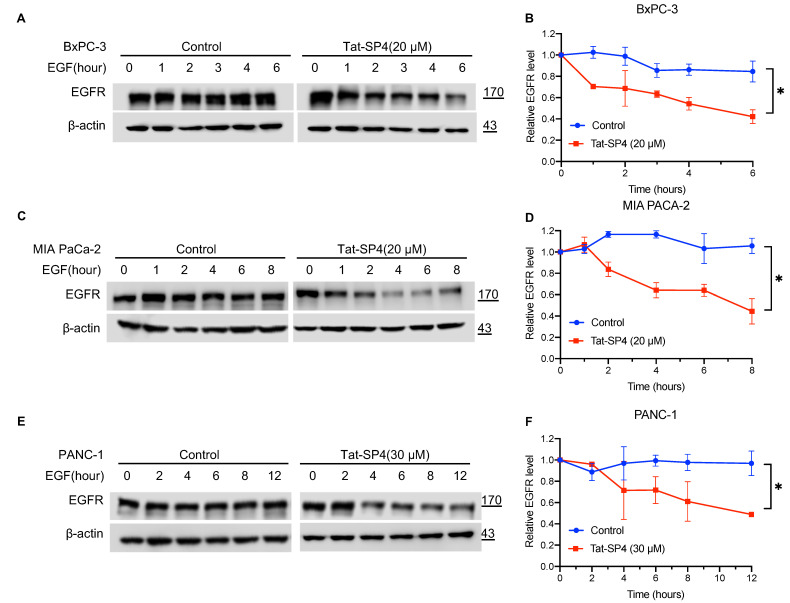
Tat-SP4 promoted endolysosomal degradation in PDAC cells. (**A**) Western blot to assess the EGFR levels in BxPC-3 cells. Cells were starved overnight and were treated with 200 ng/mL EGF together with vehicle control or 20 μM Tat-SP4 for the indicated times. (**B**) Time-dependent plots to quantify the EGFR degradation profile in BxPC-3 cells (**A**). (**C**,**E**) Similar Western blots as (**A**), but cells are MIA PaCa-2 and PANC-1, respectively. (**D**,**F**) Time-dependent plots to quantify the EGFR degradation profile in MIA PaCa-2 (**C**) and PANC-1 (**E**), respectively. Data are presented as mean ± SEM from three independent experiments; * *p* < 0.05. For Original Western Blots, see Appendix A.

**Figure 3 cancers-15-00953-f003:**
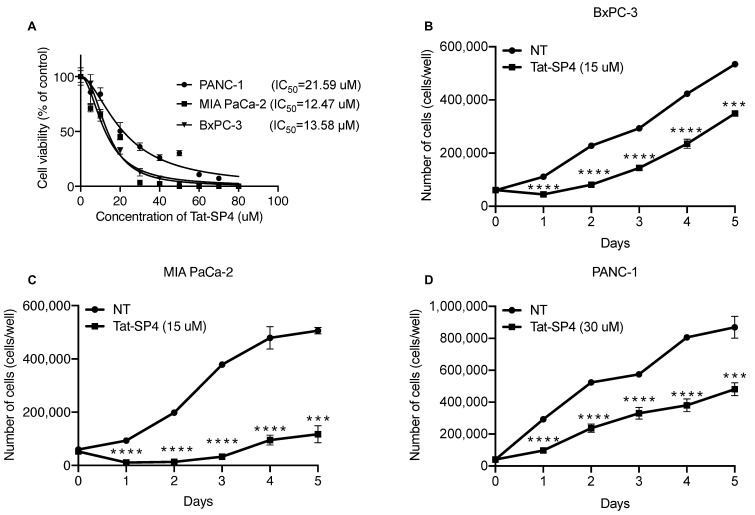
Tat-SP4 exerts potent anti-proliferative effect on PDAC cell lines. (**A**) Trypan blue exclusion assay to assess the cytotoxicity IC_50_ of the indicated peptides in PDAC cells. Cells were treated with various concentrations of the indicated peptides for 24 h. Cell numbers were manually counted by the trypan blue dye exclusion method using a hemocytometer. Data are presented as mean ± SD from three independent experiments. (**B**–**D**) Cell proliferation assay was performed over a time span of five days in PDAC cells after treatment with the indicated peptides. Cells were treated with the indicated peptides at day 0, and the cell number was calculated at the given time points (NT: no-treatment). The assay was performed in 24-well plates with three independent measurements. Data are presented as mean ± SEM from three independent experiments; *** *p* < 0.001, **** *p* < 0.0001.

**Figure 4 cancers-15-00953-f004:**
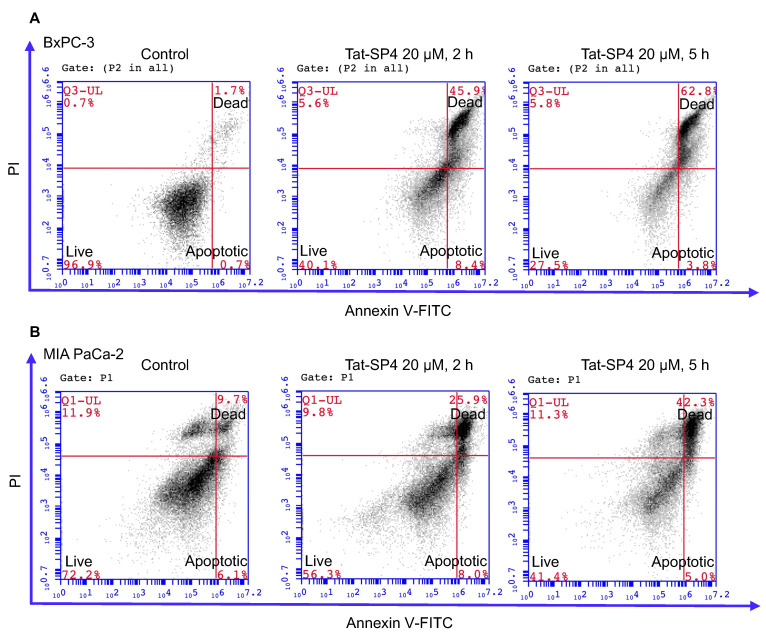
Tat-SP4 induced predominantly non-apoptotic cell death. Flow cytometry analysis with Annexin V FITC-PI staining was performed in BxPC-3 cells (**A**), MIA PaCa-2 (**B**) and PANC-1 (**C**) cells, respectively, after treatment of the indicated peptides for the indicated times. The percentage of dead cells (PI positive) in Tat-SP4 treatment groups was significantly increased compared with that of control, without notable changes in the population of apoptotic cells.

**Figure 5 cancers-15-00953-f005:**
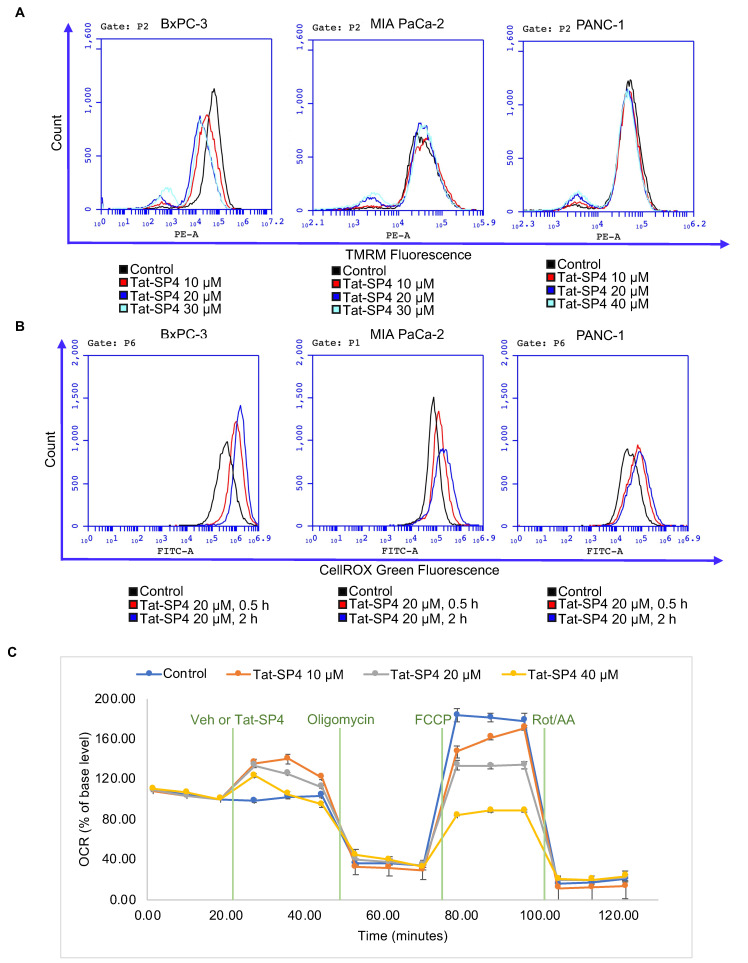
Tat-SP4 impairs mitochondria membrane potential and oxidative phosphorylation activity. (**A**) Representative flow cytometry plot of PDAC cells with the treatment of indicated concentrations of Tat-SP4 for 1 h. Cells were stained with TMRM for 30 min and analyzed in flow cytometry. (**B**) Representative flow cytometry plot of the changes in fluorescence intensity of DCF, an indicator dye of cellular ROS, under the treatment of indicated concentrations of Tat-SP4 for 30 min and 2 h, respectively. Fluorescence intensity is directly proportional to the amount of ROS species in the cell. (**C**) The changes in OCR were measured by Agilent Seahorse Analyzer using Mito Stress Test. BxPC-3 cells were treated with indicated concentrations (10 μM, 20 μM, 40 μM) of Tat-SP4 or vehicle (control) followed by sequential injection of 1 μM oligomycin, 1 μM FCCP, and 0.5 μM rotenone and antimycin A. Each data point represented the relative OCR compared to the base level. Data are presented as mean ± SEM (*n* = 5). (**D**,**E**) Quantification of basal respiration after peptide treatment and maximum respiration changes in (**C**), * *p* < 0.05, **** *p* < 0.0001.

**Figure 6 cancers-15-00953-f006:**
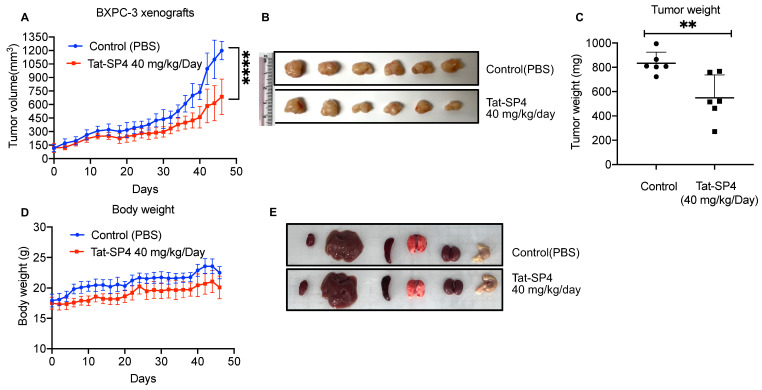
Tat-SP4 inhibited PDAC tumor growth in a xenograft model. (**A**) BxPC-3 cells were subcutaneously injected into nude mice in a number of 5 × 10^6^. The administration of 40 mg/kg Tat-SP4 or PBS (control) was started when the tumor volume reached 100 mm^3^. The tumor volume was recorded from the initial treatment to tumor harvest (Day 46). The tumor volume was defined as width^2^ × length/2. Data are presented as mean ± SEM; *n* = 6; **** *p* < 0.0001. (**B**) Picture of the dissected BPxC-3 tumor tissues from each mouse. (**C**) The tumor weight was measured after collection on day 46. Data are presented as mean ± SEM; *n* = 6; ** *p* < 0.01. (**D**) The development of body weight in mice after receiving the treatment of 40 mg/kg Tat-SP4 or PBS. Data are presented as mean ± SEM; *n* = 6. (**E**) The representative pictures of harvested vital organs from both control and treatment groups.

## Data Availability

The data presented in this study are available in the paper. Additional data related to this paper are available on request from the corresponding author.

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
