# Peer review of "Perturbation of Autophagy by a Beclin 1-Targeting Stapled Peptide Induces Mitochondria Stress and Inhibits Proliferation of Pancreatic Cancer Cells"

_cancers, 2023, doi:10.3390/cancers15030953_

Round 1

Reviewer 1 Report

This manuscript has been well designed, however before accepting author has to clarify the questions raised here

1, why author has focussed on CQ instead of other autophagy inhibitors like  3-Methyladenine (3-MA), LY294002, and Bafilomycin A1 (Baf A1) are common autophagy inhibitors that function in early autophagy by PI3K inhibition and in late autophagy by blocking vacuolar-type H(+)-ATPase. 

2,  why there is an increase in necrotic cells ? any necrotic markers can be checked?

Author Response

1, why author has focussed on CQ instead of other autophagy inhibitors like 3-Methyladenine (3-MA), LY294002, and Bafilomycin A1 (Baf A1) are common autophagy inhibitors that function in early autophagy by PI3K inhibition and in late autophagy by blocking vacuolar-type H(+)-ATPase.

CQ is one of the most used autophagy inhibitors. CQ blocks autophagy by interfering with lysosome acidification, thus sharing the same mechanism as Baf A1. A large body of literature in the autophagy field has shown that CQ, 3-MA, and Baf A1 exert a comparable effect on autophagy. Based on this knowledge, we decided to use CQ as the representative autophagy inhibitor in our study.

2, why there is an increase in necrotic cells? any necrotic markers can be checked?

Indeed our flow cytometry analysis showed that, after Tat-SP4 treatment, a significant portion of PDAC cells stained positive for both annexin V and propidium iodide (PI), which indicates necrotic cell death. As we elaborated in the Discussion section, Tat-SP4 may induce autosis, a unique form of cell death caused by excessive autophagy and marked by necrotic features such as focal rupture of the plasma membrane. The molecular mechanism of autosis is still at the early stage of investigation and no reliable markers have been reported. Future studies are needed to further characterize possible autosis in Tat-SP4-treated cancer cells.

Reviewer 2 Report

In this manuscript, Li et al evaluated the effect of a Beclin 1-targeting stapled peptide on 3 different PDAC cell lines. They found this peptide can further enhance the autophagy process, trigger faster degradation of EGFR, and induce mitochondria stress. This peptide also showed an anti-proliferative effect in PDAC cell lines and inhibited PDAC tumor growth in a xenograft mode.

Overall, the study is well-designed, the experiments are carefully performed, and the data is clearly presented. Some comments are: 

1.       The conclusion on line 233, comparing the baseline levels of LC3-II and p62 among the 3 cell lines, needs to be quantified, preferentially from the same gel.

2.       The legend of Figure 1 mentioned “individual points”, but none of the bar plots actually show individual observations.

3.       In section 3.4, for the cell death assay, data on MIA PAVA-2 cell was not shown. Please show this piece of data even if it is negative since it will still be helpful for the community.

4.       Figure 4 and Figure 5 have low resolution, and axis labels are not readable. In Figure 5B, the color used in the legend seems to be mismatched.

5.       On line 225, “increase” should be “decrease.”

Author Response

1. The conclusion on line 233, comparing the baseline levels of LC3-II and p62 among the 3 cell lines, needs to be quantified, preferentially from the same gel.

We appreciate this helpful comment. Indeed, we didn’t run the samples of the 3 cell lines on the same gel. On the other hand, BxPC-3 and PANC-1 cells have been reported to have high levels of basal autophagy activity based on the studies by Yang et al. (Genes & Development, 2011). Our data showed that Tat-SP4 enhanced autophagy furtherly in these cell lines. We have revised the manuscript and removed the statement that compared the baseline levels of LC3-II and p62 among the three cell lines.

2. The legend of Figure 1 mentioned “individual points”, but none of the bar plots actually show individual observations.

We apologize for this oversight. The manuscript has been revised accordingly.

3. In section 3.4, for the cell death assay, data on MIA PAVA-2 cell was not shown. Please show this piece of data even if it is negative since it will still be helpful for the community.

We thank the reviewer for this helpful comment. We performed the relevant experiment and added the data to Figure 4B. Our data shows that Tat-SP4 induced predominantly non-apoptotic cell death in MIA PaCa-2, similar to what we observed for BxPC-3 and PANC-1,

4. Figure 4 and Figure 5 have low resolution, and axis labels are not readable. In Figure 5B, the color used in the legend seems to be mismatched.

We apologize for our oversight. The figures have been updated with high-resolution versions and the color reference in the legend of Figure 5B has been revised as well.

5. On line 225, “increase” should be “decrease.”

We apologize for our typo. The manuscript has been revised.

Reviewer 3 Report

 In the manuscript “Perturbation of autophagy by a Beclin 1-targeting stapled peptide induces mitochondria stress and inhibits proliferation of pancreatic cancer cells” by Li et al., the author has shown that the Beclin 1-targeting stapled peptide (Tat-SP4) enhanced autophagy in multiple PDAC cell lines. Tat-SP4 also triggered faster endolysosomal degradation of EGFR and induced significant mitochondria stress. Tat-SP4 exerted a potent anti-proliferative effect in PDAC cell lines in vitro and prohibited xenograft tumor growth in vivo. The findings in the manuscript are well presented and identify a relevant pathway. However, there are a few concerns that the authors must address before publication.

1.      Tat-SP4 is a Beclin 1-targeting stapled peptide (autophagy inhibitor). Then why the author has not measured the level of Beclin-1 (an essential autophagy protein) in any of the PDAC cell lines after treatment with Tat-SP4?

2.      What is the molecular mechanism of Tat-SP4-induced anti-proliferative effect PDAC cell lines (what is the major pathway through which Tat-SP4 is mediating its anti-proliferative effect)?

3.      What is the effect of Tat-SP4 on different phases of the cell cycle?

4.      Why has the author not measured the LC3 and p62 levels in xenograft tumor tissue by IHC?

5.      What is the author's point of view in terms of the effect of Tat-SP4 on PDAC cancer stem cells?

6.      Why the author has not measured the formation of autophagosomes in Tat-SP4 treated PDAC cells by immunofluorescence (e.g., using dye monodansylcadaverine (MDC))

7.      Why the author has not checked the molecular mechanism of Tat-SP4-induced autophagy in PDAC cell lines?

In addition, there are some typos;

Figure 3: NT referrers to what? Mention in the figure legend

Line no. 163- rewrite “apoptosis in PDA cells” as “apoptosis in PDAC cells”

Author Response

1. Tat-SP4 is a Beclin 1-targeting stapled peptide (autophagy inhibitor). Then why the author has not measured the level of Beclin-1 (an essential autophagy protein) in any of the PDAC cell lines after treatment with Tat-SP4?

We thank the reviewer for this interesting question. The design principle for our Beclin 1-targeting stapled peptides is to disrupt the functionally inactive Beclin 1 homodimer and promote the formation of functionally active Beclin 1-Atg14L/UVRAG heterodimer to induce autophagy. Thus Tat-SP4 is not expected to affect the overall level of Beclin 1 in vivo but to promote the homodimer-to-heterodimer transition instead. In our previous study (Wu et al., PNAS, 2018), we did Co-IP experiments and confirmed that Tat-SP4 indeed did not affect the overall level of Beclin 1 in HEK293 cells but triggered homodimer-to-heterodimer transition as expected. Tat-SP4 likely induces autophagy in the same manner in all Beclin 1-expressing cell lines. Thus we wouldn’t expect Tat-SP4 to affect the level of Beclin 1 in PDAC cells. As a result, we only measured autophagy markers LC3 and p62.

2. What is the molecular mechanism of Tat-SP4-induced anti-proliferative effect PDAC cell lines.(what is the major pathway through which Tat-SP4 is mediating its anti-proliferative effect)?

Our conclusion is Tat-SP4 induced non-apoptotic death in PDAC cells to inhibit the overall proliferation. W proposed that Tat-SP4 might induce autosis, a unique form of cell death that is correlated to excessive autophagy. Future studies are needed to identify the molecular pathways and machineries that execute autosis.

3. What is the effect of Tat-SP4 on different phases of the cell cycle?

Autophagy has been found to participate in many aspects of cell cycle regulation (Beth Levine et al., Developmental cell, 2008). Based on our observation, Tat-SP4 induced non-apoptotic cell death within hours without direct effect on the cell cycle.

4. Why has the author not measured the LC3 and p62 levels in xenograft tumor tissue by IHC?

We thank the reviewer for this inquisitive comment. We expect that the levels of LC3 and p62 in xenograft tumor tissue would be similar to that in BxPC3 cells cultured in medium.  If we want to investigate whether xenograft tumor cells have altered autophagic response, then doing such assay would require culturing of the primary tumor sample to measure the autophagic response to stress. We will definitely explore this assay in future studies.

5. What is the author's point of view in terms of the effect of Tat-SP4 on PDAC cancer stem cells?

We thank the reviewer for this insightful comment. Autophagy has been implicated in the generation, differentiation, plasticity, and migration/invasion of cancer stem cells. (Nazio et al., cell death and differentiation, 2019). We have updated the Discussion section to include this point.

6. Why the author has not measured the formation of autophagosomes in Tat-SP4 treated PDAC cells by immunofluorescence (e.g., using dye monodansylcadaverine (MDC))

Our previous study (Wu et. al. PNAS 2018) showed that Tat-SP4 induced the formation of more autophagosomes in HEK293 cells as measured by confocal immunofluorescence microscopy. Tat-SP4 likely induces autophagy in the same manner in PDAC cells. As a result, we decided that measuring levels of LC3 and p62 to track autophagy activity would be sufficient.

7. Why the author has not checked the molecular mechanism of Tat-SP4-induced autophagy in PDAC cell lines?

As we stated in #1, the molecular mechanism of Tat-SP4 induced autophagy was thoroughly investigated in our previous study (Wu et al., PNAS, 2018). As a result, we didn’t carry out the same experiments in PDAC cells.

In addition, there are some typos;

Figure 3: NT referrers to what? Mention in the figure legend

Line no. 163- rewrite “apoptosis in PDA cells” as “apoptosis in PDAC cells”

We apologize for our mistake. NT referrers to no-treatment. We have mentioned it in the figure legend.